# Cellular lensing and near infrared fluorescent nanosensor arrays to enable chemical efflux cytometry

Soo-Yeon Cho [1], Xun Gong[1], Volodymyr B. Koman [1], Matthias Kuehne [1], Sun Jin Moon[1], Manki Son[1], Tedrick Thomas Salim Lew [1,2], Pavlo Gordiichuk[1], Xiaojia Jin [1], Hadley D. Sikes [1] & Michael S. Strano [1✉]

Nanosensors have proven to be powerful tools to monitor single cells, achieving spatio-temporal precision even at molecular level. However, there has not been way of extending this approach to statistically relevant numbers of living cells. Herein, we design and fabricate nanosensor array in microfluidics that addresses this limitation, creating a Nanosensor Chemical Cytometry (NCC). nIR fluorescent carbon nanotube array is integrated along microfluidic channel through which flowing cells is guided. We can utilize the flowing cell itself as highly informative Gaussian lenses projecting nIR profiles and extract rich information. This unique biophotonic waveguide allows for quantified cross-correlation of biomolecular information with various physical properties and creates label-free chemical cytometer for cellular heterogeneity measurement. As an example, the NCC can profile the immune heterogeneities of human monocyte populations at attomolar sensitivity in completely non-destructive and real-time manner with rate of ~600 cells/hr, highest range demonstrated to date for state-of-the-art chemical cytometry.

[1] Department of Chemical Engineering, Massachusetts Institute of Technology, Cambridge, MA, USA. [2] Institute of Materials Research and Engineering (IMRE), Agency for Science, Technology and Research (A*STAR), Singapore, Singapore. ✉email: strano@mit.edu

Nanotechnology has produced some of the most sensitive analytical platforms for molecules in existence, with many achieving single-molecule resolution[1–3], including arrays for DNA sequencing[4,5] as well as reactive oxygen species (ROS) detection[6,7]. There is significant interest and motivation to extend such platforms to the study of living cells[8,9] and microbes[10,11], where they can form the basis of non-destructive techniques to probe various biochemical mechanisms. This has obvious applications to medicine and life science research and of particular importance to the emerging area of cell-based therapies and regenerative medicine for the treatment of cancer, leukemia, and neurodegenerative diseases[12–14]. However, cellular populations are necessarily heterogeneous, and cellular therapies necessarily require characterization methods that are non-destructive and do not contaminate the cells themselves[15], ruling out conventional flow cytometry that requires fluorescent labels[16]. Extending various types of nanosensors to statistically relevant numbers of living cells and organisms in a non-destructive manner remains unaddressed to date with the basic problem of nanosensors including interfacing strategy, signal-transducing mechanism, and mechanical robustness[17].

Various label-free cell imaging techniques such as digital holographic microscopy (DHM)[18–20] or optical diffraction tomography[21–23] have been developed for high-throughput cell classification based on image analysis. For example, Ugele et al. discriminated against healthy and pathological blood cells using holographic speckle images of DHM technique[18]. Singh et al. used machine learning-based hologram screening to detect tumor cells in high-throughput[19]. However, these techniques are based on physical property measurements from cell images. Chemically quantification for heterogeneity in cell populations is still an open problem. Flow and chemical cytometry has been widely used to quantify the molecular heterogeneities of target cell populations. While typical flow and image cytometry of living cells can sample $10^6$–$10^7$ cells in just a few minutes[24–26], the state of the art for the emerging field of chemical cytometry is between 50 and 500 cells/h since cells need to be pre-labeled, lysed, and separated to be detected[27–29]. Nevertheless, this level of throughput has elevated chemical cytometry as a valuable cell characterization tool allowing quantitative information to be gathered with high selectivity and signal-to-noise ratio[30,31]. Nanosensors have significant potential to greatly expand the number of variables measured in chemical cytometry given the large number of types being demonstrated in the recent literature[32–36]. Organic and inorganic fluorescent nanoparticles have been used to monitor intra- and extracellular information of single cells successfully[34–36]. Near-infrared (nIR) fluorescent single-walled carbon nanotubes (SWNT) are particularly promising components toward label-free and single-molecule level cellular profiling. To date, they have been developed for the detection of single-cell biochemical efflux for antibodies, neurotransmitters, and ROS[9,37–40]. Additionally, their rapid and direct optical readout is ideal for sensor interfacing, and carbon, in particular, possesses photostability, biocompatibility, and tunable chemical selectivity for this purpose[17,40–42].

In this work, we developed a nanosensor chemical cytometry (NCC) that can characterize the real-time chemical efflux of cell populations at high throughput. nIR fluorescent SWNT nanosensors are uniformly integrated within a cell-transporting microfluidic channel. Each single cell optically interacts with the underlying nanosensor array, producing an informative nIR optical lensing profile that can be modeled as a photonic nanojet. Within this biophotonic waveguide, cells can be both visualized and chemically tracked in real-time and at high resolution, without the need for labeling or additional optical manipulation. Based on the combination of nanosensor response and observed cellular lensing properties, the NCC platform is able to yield multivariate data that inform the heterogeneities of human monocyte populations at the attomolar ($10^{-18}$ moles) level of $H_2O_2$ efflux. Furthermore, this type of cellular population data allows for phenotypic correlation between real-time chemical efflux and various biophysical properties of each individual cell including diameter, eccentricity, and refractive index (RI).

## Results and discussion

**Nanosensor integration with microfluidics.** The schematic of the flow channel and nanosensor array integration for NCC are shown in Fig. 1a. The array is demonstrated using a $(GT)_{15}$ DNA wrapped SWNT (SWNT/$(GT)_{15}$), which was previously shown to exhibit nIR intensity attenuation upon selective detection of $H_2O_2$[7,42]. $H_2O_2$ efflux was targeted for the application due to its central role in cellular signaling and immune responses[6,9]. For the first step, a micro-droplet of (3-aminopropyl) triethoxysilane (APTES) was injected into a pristine channel and incubated. A commercial microfluidic channel was coated with APTES for self-assembled monolayer formation and SWNT/$(GT)_{15}$ adhesion on both the top and bottom surface of the channel. Subsequently, the channel was washed with phosphate buffer saline (PBS) and a micro-droplet of SWNT/$(GT)_{15}$ dispersion was injected into the channel. Stable dispersions of nanosensors were confirmed via UV–vis–nIR absorption spectra of SWNT/$(GT)_{15}$ (Supplementary Fig. 1). During evaporation, nanosensor particles necessarily align at the three-phase line of the micro-droplet pinned at the end of the flow channel (Fig. 1b). This resulted in a uniform array on both top and bottom surfaces of the channel following the evaporation-induced self-assembly (EISA)[43]. After the EISA, the channel was flushed with PBS again to remove unbounded residual nanoparticles. Completed nanosensor integrated microfluidics (NIMs) were highly transparent to visible light indicating an absence of aggregation or large array defects (Fig. 1c). Polarized Raman spectroscopy of NIM showed the depolarization ratio of 0.61 from $G$ band intensity demonstrating that the nanosensors were aligned along the flowing direction of the channel during EISA (Fig. 1d)[44]. nIR imaging was used to investigate the fluorescence signal mapping of the NIM (Fig. 1e). While we find NIMs to display strong nIR fluorescence, uncoated channels show no nIR signal (Supplementary Fig. 2). In addition, NIM without APTES treatment showed severe nanosensor aggregation during the EISA process and consequently, nanosensors were completely removed with PBS flowing, indicating that surface chemistry of the microfluidic channel is critically important to uniform and stable EISA process. A magnified nIR image of NIM with single-cell size (20 μm diameter) shows that nanosensors are homogeneously and continuously deposited with ~720 local detector pixels across a single cell (Fig. 1f and Supplementary Fig. 3). Atomic force microscopy (AFM) demonstrated that nanosensor bundles were densely and homogeneously covered on the channel surface at the micron-scale (Supplementary Fig. 4). Consequently, the nanosensor array on the microfluidic channel could clearly visualize the cells flowing through the channel and maximize the signal-to-noise ratio of the signal from cell efflux for NCC[38]. As the concentration of nanosensor dispersion increases, uniformity of nanosensor array was enhanced with a significant decrease of voids, and aggregation of nanosensors and 80 mg/L coating shows the highest nIR intensity with the most uniform pixel distributions (Supplementary Fig. 5). Nanosensors were uniformly coated on the top and bottom surfaces of the channel during EISA, as shown by the comparable nIR pixel distributions along both surfaces (Fig. 1g). Peak position and relative peak intensities of nIR spectrum of NIM were almost identical with SWNT in the dispersion phase, indicating that the dielectric environment surrounding the immobilized nanosensors were similar (Fig. 1h)[45].

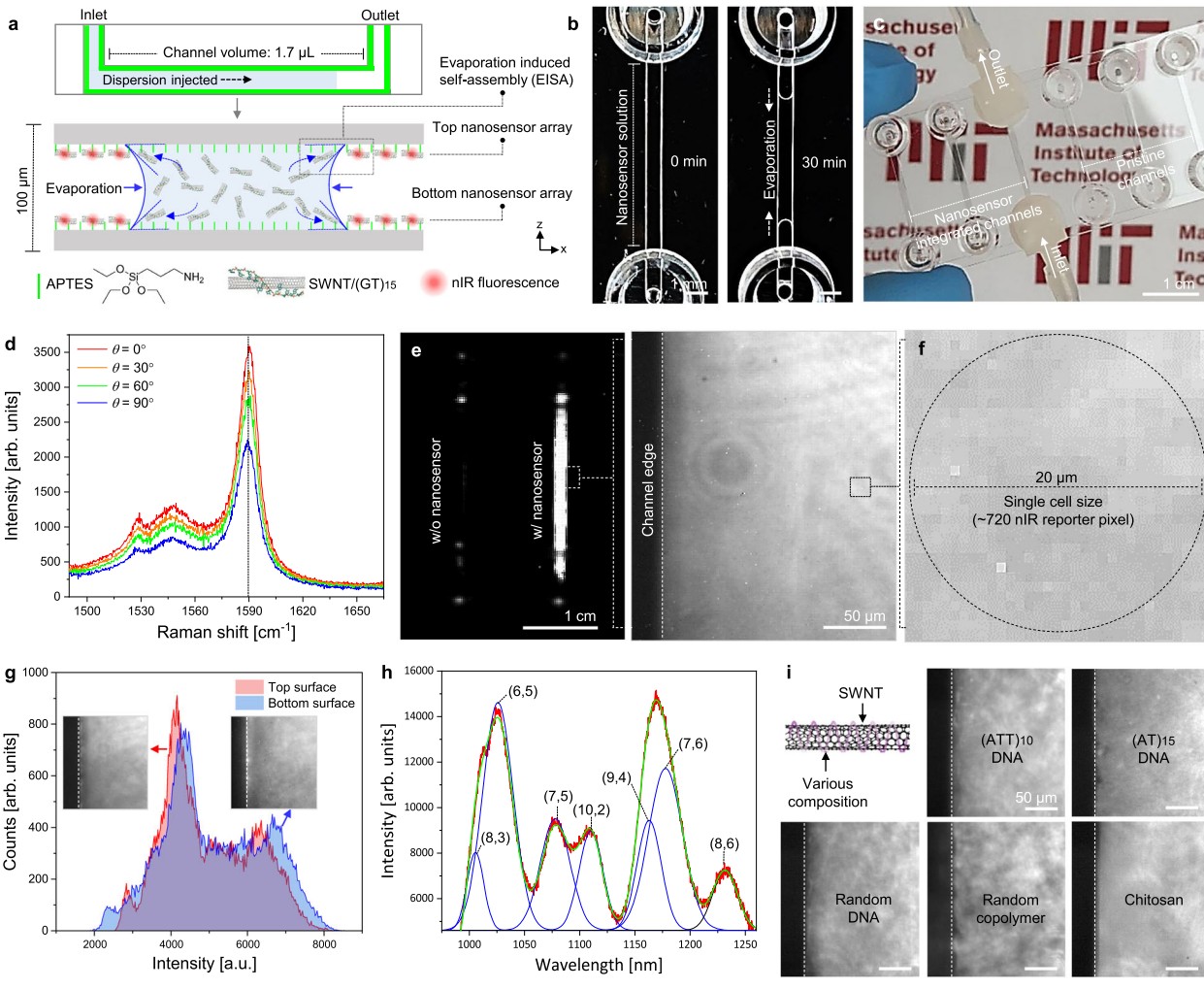

**Fig. 1 Nanosensor integration with microfluidics. a** Schematic illustration of the nanosensor integration process with microfluidics using EISA. **b** Photograph of EISA process of NIM for 0 min (left) and 30 min (right). **c** Photograph of completed multi-array NIM and pristine channel. **d** Polarized Raman spectrum (G-peak) of NIM. **e** nIR images of NIM and pristine channel. **f** Magnified nIR image of NIM with single-cell size resolution (20 μm) having ~720 nIR reporter pixel. **g** Histograms of nIR pixel intensities of top and bottom NIM surfaces NIM (inset: nIR images of the top and bottom surfaces). **h** nIR fluorescence spectrum of NIM. **i** nIR images of NIM with varying composition of SWNT nanosensor integration.

Varying compositions of SWNT nanosensors ($(GT)_{15}$ DNA, $(AT)_{15}$ DNA, $(ATT)_{10}$ DNA, random DNA, chitosan) were integrated with the microfluidic channels by our EISA-based NIM fabrication process for monitoring of various chemical components of the cell (Fig. 1i)[46].

**Chemical detection performances of NIM.** In-vitro $H_2O_2$ detection experiments were conducted to investigate the chemical sensing performance of the NIM. The fluorescence intensity from all SWNT chiralities decreased with 10–20% relative magnitude upon exposure to 1 μM $H_2O_2$ (Fig. 2a). Real-time nIR images of NIM show that the channel emission is completely quenched with 1 M $H_2O_2$ flowing (Fig. 2b). This is attributed to that $H_2O_2$ molecules selectively adsorbed on nanotube sidewall donate electrons directly to the conduction bands of SWNT/$(GT)_{15}$, and extra electrons in the conduction bands can then quench excitons through non-radiative recombination (Fig. 2c)[7,47]. Real-time nIR signals $((I - I_0)/I_0)$ were measured with a wide range concentration of $H_2O_2$ injection (Fig. 2d). Here, $I_0$ and $I$ represent the nIR intensity of the channel at $t = 0$ and after $H_2O_2$ injection, respectively. Upon $H_2O_2$ injection, the NIM showed an instantaneous and continuous decrease in nIR signal on the order of 5–80% depending on $H_2O_2$ concentration. For the first-order

reversible reaction, the relationship between the analyte and available docking sites for $H_2O_2$ can be described as follows[48]:

$$A + \theta \rightleftarrows A\theta \tag{1}$$

the equilibrium for this reaction can be modeled as

$$K_A = \frac{[A\theta]}{[A][\theta]} \tag{2}$$

Assuming that the sensor response is proportional to the $A\theta/\theta_{tot}$ ratio, it is found that

$$\frac{I - I_0}{I_0} = \alpha \frac{[A\theta]}{[\theta_{tot}]} + \beta = \alpha \frac{([A]K_A)^n}{([A]K_A)^n + 1} + \beta \tag{3}$$

with the total concentration of available recognition sites $[\theta]_{tot}$ and the parameter $n$ for cooperativity. Fitting the data in Fig. 2e with Eq. (3) ($R^2 = 0.9983$) results in a proportionality factor $\alpha = 88.74$ with $\beta = 2.30$, $K_D = 1/K_A = 0.00204$ M, and $n = 0.317$, indicating negative cooperativity in good agreement with previous papers ($n < 1$)[42,45,48]. The limit of detection in this mode is 11.56 nM; this value was calculated by adding the NIM sensor response from the addition of only buffer (PBS) to 3-times the standard deviation ($\sigma$). A response time of <9 min was achieved based on the time it takes to reach 90% value of the minimum nIR level (Fig. 2f).

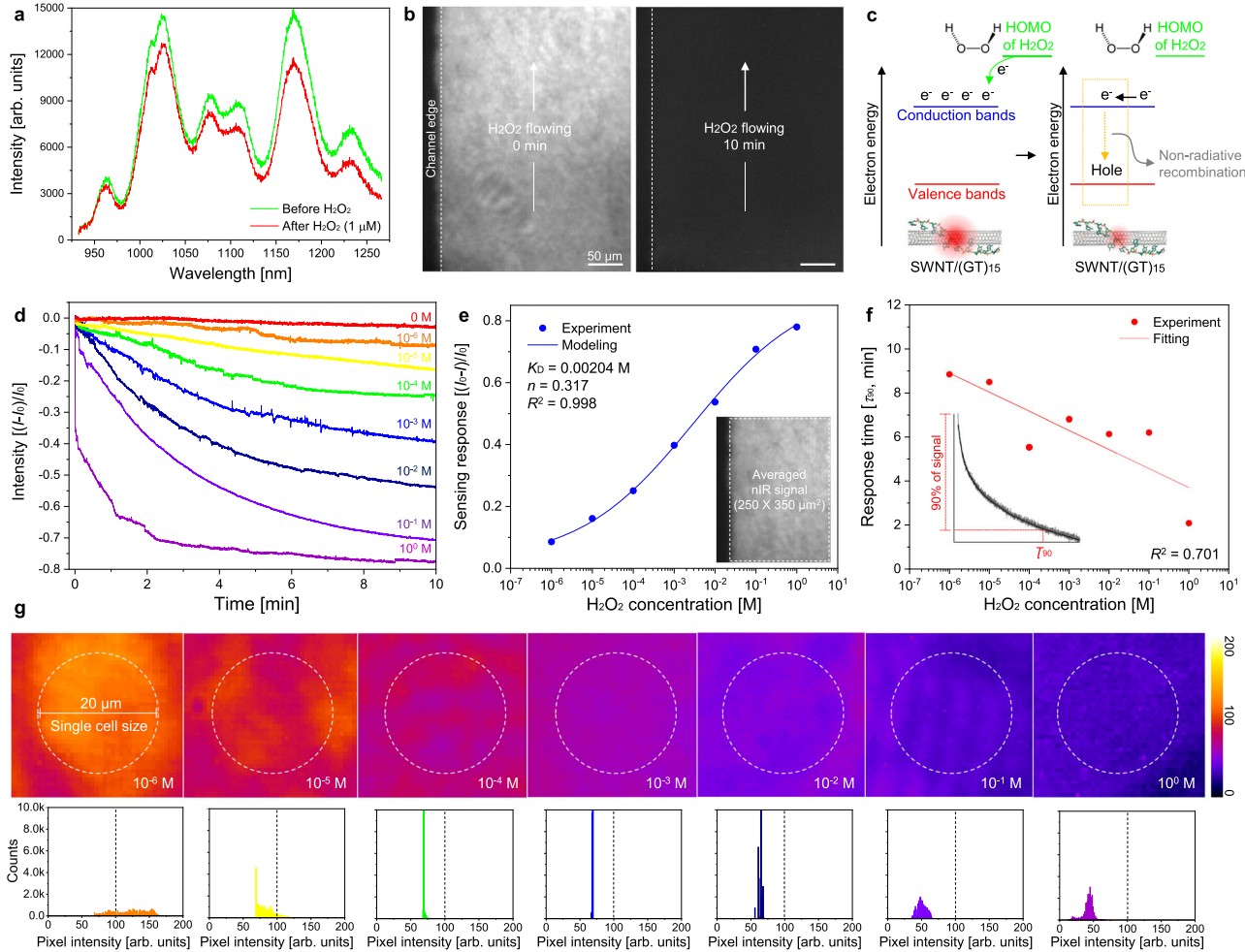

**Fig. 2 In vitro chemical detection performances of NIM. a** nIR spectrum of NIM with $H_2O_2$ solution flowing (1 μM, 1 μL/min). **b** nIR images of NIM before and after $H_2O_2$ flowing (1 M, 10 μL/min, 10 min). **c** Schematic illustration of $H_2O_2$ detection mechanism of SWNT/$(GT)_{15}$ nanosensor. **d** Real-time nIR response of NIM with various concentrations ($10^{-6}$, $10^{-5}$, $10^{-4}$, $10^{-3}$, $10^{-2}$, $10^{-1}$, $10^0$ M) of $H_2O_2$ injection (10 min). **e** Maximum response amplitude and **f** response time of NIM with various concentrations of $H_2O_2$. The data represent the mean value of 250 × 350 μm² NIM measurement. **g** nIR snapshots and intensity histogram (fire scale, ImageJ) of NIM with single-cell size resolution (20 μm) after 10 min flowing of various concentrations of $H_2O_2$.

The NIM platform demonstrates uniform and near-instantaneous nIR intensity response even when imaged at the high-resolution needed to interrogate single cells (~20 μm) (Fig. 2g).

**Cellular lensing effect**. For the NCC implementation, the NIM was integrated with a syringe pump and nIR microscope. 561 nm excitation laser was provided from the bottom side of the channel (right, Fig. 3a). Human monocytes (U937) were cultured as chemical cytometry targets (Supplementary Fig. 6) since they are widely studied in biomedical fields with heterogeneous differentiation behavior into macrophages by immune activation[49,50]. This monocyte-derived macrophage exhibits distinct ROS efflux in real-time as an immune response to various kinds of infection/inflammation. Measuring subtle molecular differences of ROS efflux can also benefit the detection and prevention of cardiovascular disease and neurodegenerative disorders[51,52]. Therefore, a tool that would enable the precise profiling of the dynamic antigenic response of single monocyte and eventually immune heterogeneities as a function of different cellular physical properties could lead to mechanistic understanding and therapeutic development for these conditions. We found that the flowing cells optically interact with the underlying nanosensor emitter array and create a moving, label-free region of the highest sensor signal by lensing the photoemission through the flowing cell itself

(Fig. 3b, c and Supplementary Movie 1). This cell visualization was directly affected by both the uniformity and intensity of the underlying nanosensor array (Supplementary Fig. 7). A magnified nIR image of a single flowing cell shows that the contour and shape of the monocyte could be visualized as observed in an OM (inset, top-right) with the highest nIR intensity from the nanosensor array corresponding to the center, and Airy rings visible around the periphery (Fig. 3c). Micro-particles larger than the illumination wavelength can similarly function as focusing lens[53,54]. When particles have a RI contrast ratio with the fluid medium <2:1 and a diameter ($d_μ$) larger than the wavelength (~$2λ < d_μ < 40λ$), a highly focused propagating beam from the shadow-side of the surface is generated due to constructive interference of the light field, called a photonic nanojet[55,56]. For our system, the nIR fluorescence ($λ$: 1–1.25 μm) from the top nanosensor array passes through the membranes, cytoplasm, and nucleus of the underlying flowing cells of mean diameter 10–20 μm. The estimated RI of the cell components are $n_n = 1.43 ± 0.04$ for the monocyte nucleus, $n_c = 1.348 ± 0.004$ for the monocyte cytoplasm (average cell = 1.383), and $n_m = 1.33$ for the flowing media, which are optimum optical conditions for the photonic nanojet effect ($n_{cell}/n_m = 1.039$ (<2))[57,58]. Consequently, nIR photoemission from the integrated nanosensor array was refracted through the flowing cell and focused at a certain focal point

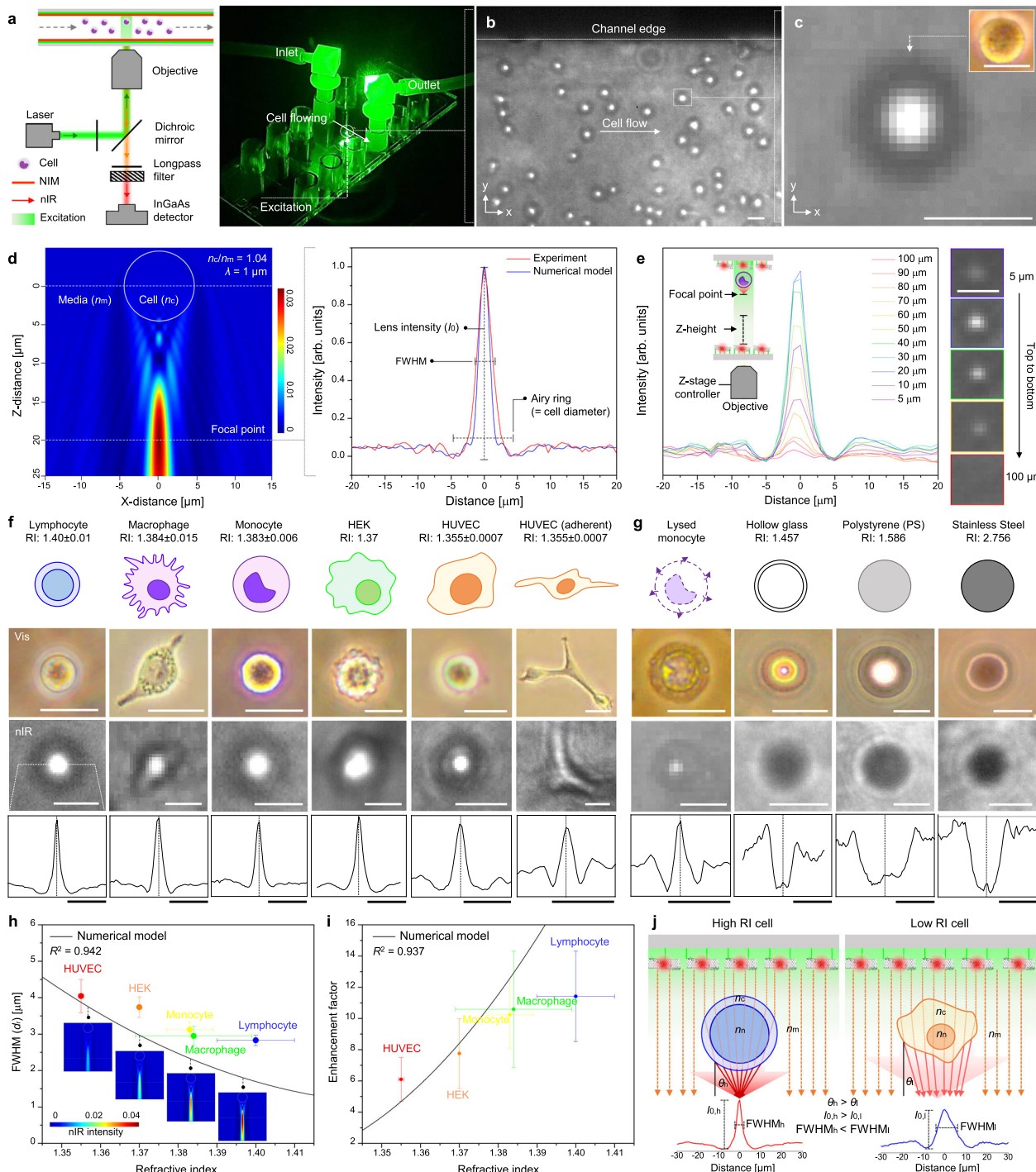

**Fig. 3 Cellular lensing effect. a** Instrumental setup for NCC implementation: schematic illustration (left) and a photograph (right). **b** nIR images of human monocytes flowing (0.5 μL/min) NIM. **c** Magnified nIR image of single monocyte in NIM (inset: OM image of single monocyte). **d** FDTD numerical modeling for photonic nanojet and fitting with experimental cellular lensing profile ($n_c/n_m = 1.04$, $\lambda = 1$ μm). **e** nIR lensing profiles of a single cell with various focusing points from 5 to 100 μm along Z-stage. nIR lensing effects of **f** various live cells and **g** reference micro-particles (top-to-bottom: schematics, OM, nIR images, lensing profiles). **h** FWHM and **i** enhancement factors of various cells with numerical model. Data are mean (circle) ± σ (error bar), with $n_{cell} = 10$. **j** Schematic illustrations for different lensing behavior of a high RI cell (left) and a low RI cell (right). Scale bars: 20 μm.

below it, a phenomenon called cellular lensing. Several previous papers reported such photonic nanojet based micro-lensing behavior of cells in visible spectrum ($\lambda$: 0.473–0.644 μm) using optical tweezing[59–62]. In this study, we observe the photonic nanojet phenomena through a flowing cell itself in the nIR range ($\lambda$: 1–1.25 μm) by exploiting nanosensor array as an illumination source. Based on this, we could correlate both biochemical and

biophysical properties of live cells with cellular lensing profiles. Consequent nIR intensity profiles of single cell showed the highest lens intensity ($I_0$) with 3–5 μm full width at half-maximum (FWHM) and following Airy rings corresponding to the cell diameter (red plot in Fig. 3d). This nIR lensing profile was measured for multiple cells ($n = 20$) with almost identical FWHM (3.37 μm, $\sigma = 0.22$) and enhancement factor (9.43,

$\sigma = 1.86$), indicating that this lensing effect is reliable and specific to certain cell status (Supplementary Fig. 9).

Finite-difference time-domain (FDTD) numerical modeling can demonstrate the cellular lensing as originating from a photonic nanojet effect[55,56]. Micro-spherical structures having similar diameters between 10 and 20 μm, eccentricity, and RI values ($n_c/n_m = 1.04$) compared with cells were used as targets for FDTD modeling. The spherical target is excited by an incident plane wave of wavelength 1 μm corresponding to the fluorescence emission of the nanosensor array (modeling details in the "Methods" section and Supporting Information). The resulting optical intensity distribution map shows that light from top side of the target strongly focuses at a 20 μm distant point from the center of the cell forming a 2–4 μm wide light jet (left, Fig. 3d). The model describes the experimental light intensity profile of the cellular nanojet at 20 μm focal distance with high fidelity in terms of $I_0$, FWHM, including Airy rings (right, Fig. 3d). We note a slight deviation between the FDTD model and experiment for the Airy rings and FWHM possibly originating from the non-uniformity of nIR excitation source and ellipticity of the monocytes. When the excitation light was focused on the bottom surface of the NIM (at Z-stage = 100 μm), the target cell is not distinguishable above the background (red line) (Fig. 3e and Supplementary Fig. 10). A slight lensing peak ($I_0$) begins to be observed at 80 μm (orange line), and is highest in intensity at 20–30 μm distance from the top surface with an enhancement factor of 9.1 (blue line), in agreement with the focal points of the FDTD numerical model. The variation in lensing intensity as a function of focusing distance also shows excellent agreement between model and experiment (Supplementary Fig. 16). This agreement gives confidence that cellular lensing images are indeed projected 20 μm from the cell center and therefore observable for those flowing within 10 μm of the NIM top surface.

This unique nIR lensing effect was not only observed for monocytes, but also for other type of cells including lymphocyte, macrophages, epithelial cells (e.g. human embryonic kidney cells (HEK)), and endothelial cells (e.g. human umbilical vein endothelial cells (HUVEC)) (Fig. 3f). Since all the cells are composed of cytoplasm, nucleus, and membrane[58], which have higher RI than that of media ($n_m$) but a ratio under 2, all cell species could form photonic nanojet and nIR lensing effect following their own shape and contour. Even cells that adhere on the channel surface such as HUVEC apparently display the profile of nIR lensing albeit with weaker intensity than suspended cells due to the smaller thickness (Supplementary Fig. 11). In contrast, reference micro-particles similar in size with cells of interest between 15 and 25 μm such as glass spheres, polystyrene (PS), and stainless steel particles with higher RI than $n_m$ (1.457, 1.586, and 2.756, respectively) display no nIR lensing. Note that the nIR fluorescence is highly refracted or reflected on surfaces and over-focused within such reference particles due to the high RI values (Fig. 3g)[55]. In addition, we observe significantly weaker cellular lensing with lysed monocytes. In this case, the absence of cytoplasmic content reduces the RI to close to $n_m \sim 1.33$, inhibiting nIR refraction. Accordingly, the observed nIR lensing effect appears to be a phenomenon unique to live cells having optimal RI, diameter, and composition for the formation of a nIR photonic nanojet.

Distinct nIR lensing profiles were observed for each cell type corresponding to unique RI ranges ($1.40 \pm 0.01$, $1.384 \pm 0.015$, $1.383 \pm 0.006$, $1.37$, and $1.355 \pm 0.0007$, for B lymphocyte[63], macrophage[64], monocyte[57], HEK[65], and HUVEC[66], respectively; RI values are those reported previously). The FWHM and enhancement factor of each cell can be calculated and described with an FDTD numerical model (Fig. 3h, i, respectively) with good agreement ($R^2 = 0.942$ and $0.950$, respectively). Model predictions show that cellular lensing can be utilized to estimate a wide range of biophysical properties of the cell including diameter, eccentricity, and RI (Supplementary Figs. 12–15).

For example, cells with higher RI show distinctly smaller FWHM and larger enhancement factors than cells with lower RI, in agreement with the FDTD model. High RI cells such as B lymphocytes are composed of larger nucleus volumes ($n_n$) than cytoplasmic components ($n_c$) for antibody and cytokine production[67]. Thus, the nIR excitation wave becomes more refracted through a high RI cell and thus more tightly focused onto focal points compared with low RI HUVEC cells (Fig. 3j)[58]. In this way, nIR cellular lensing in this NCC platform provides a unique opportunity to cross-correlate the chemical efflux as measured by the underlying nanosensor array with distinct biophysical properties such as cell size, eccentricity, and RI, which is the most closely related physical variables to cellular signaling mechanism. Ultimately, these properties can be linked to critical attributes such as viability, metabolic, membrane mechanistic properties, or intracellular composition, quantitatively correlating them with biochemical information.

**Real-time chemical efflux detection using cellular lensing effect.** We find that when human monocytes are injected into the NIM in a controlled stopped-flow system, distinct nIR intensity variations can be observed for individual monocytes corresponding to different immune activation states (Fig. 4a). We use phorbol 12-myristate 13-acetate (PMA) to induce immune activation of the human monocytes since it is a known agonist of the protein kinase C (PKC) signaling cascade. PKC activates nicotinamide adenine dinucleotide phosphate (NADPH) oxidase and consequently stimulates $H_2O_2$ secretion during differentiation into macrophages (Supplementary Fig. 17)[68]. NADPH oxidase activity generates other ROS species including superoxide anion ($O_2 \cdot^-$) and hydroxyl radical ($OH \cdot^-$) of course but at significantly lower levels of $10^3$ and $10^8$ times less than $H_2O_2$, respectively[69,70]. It is safe to assume that $H_2O_2$ is the dominant efflux from monocyte activation. Time series nIR images show that the $I_0$ corresponding to the immune activated monocyte (+PMA) (middle of Fig. 4a) decreases relative to non-activated monocyte (−PMA) (left of Fig. 4a) with increasing time. Catalase, an enzyme that decomposes $H_2O_2$[71], suppresses the signal as a negative control (right of Fig. 4a). To analyze quantitatively, the nIR pixels corresponding to the nanosensor array were integrated for each cell and labeled ($I_{cell}$), producing three cell populations per experiment (+PMA, −PMA, and +PMA & catalase) (Fig. 4b). Activated monocytes show significant variation in their real-time nIR nanosensor response while −PMA showed slow and small variation over the 500 s measurement window. We detect a basal $H_2O_2$ level even for the non-activated monocytes without PMA activation, which is consistent with the literature[72]. As expected, +PMA & catalase showed invariant sensor responses attributed to $H_2O_2$ decomposition by the enzyme. The +PMA group ($n = 41$) had an average of 4.5- and 3.4-times higher intensity variations than −PMA and +PMA & catalase groups (Supplementary Fig. 18). Also, the nIR image of single monocytes shows distinct quenching traces after measurements consistent with a response due to $H_2O_2$ efflux (Supplementary Fig. 19a, b).

The schematic in Fig. 4c summarizes this real-time $H_2O_2$ efflux detection for single cells using the cellular lensing effect. The moving cell within the flow field exhibits strong nIR lensing from the induced photonic nanojet while the $H_2O_2$ efflux is minimal at the underlying nanosensor array (Supplementary Fig. 19c, d). During the periodic stopped-flow, the $H_2O_2$ efflux cloud surrounding each cell starts to register on the projected nanosensor area nearest to the cell, resulting in a quenching of the immediate spot. This quenching allows for precise quantification of the $H_2O_2$ efflux. At this point, the nIR lensing power is drastically reduced with weaker

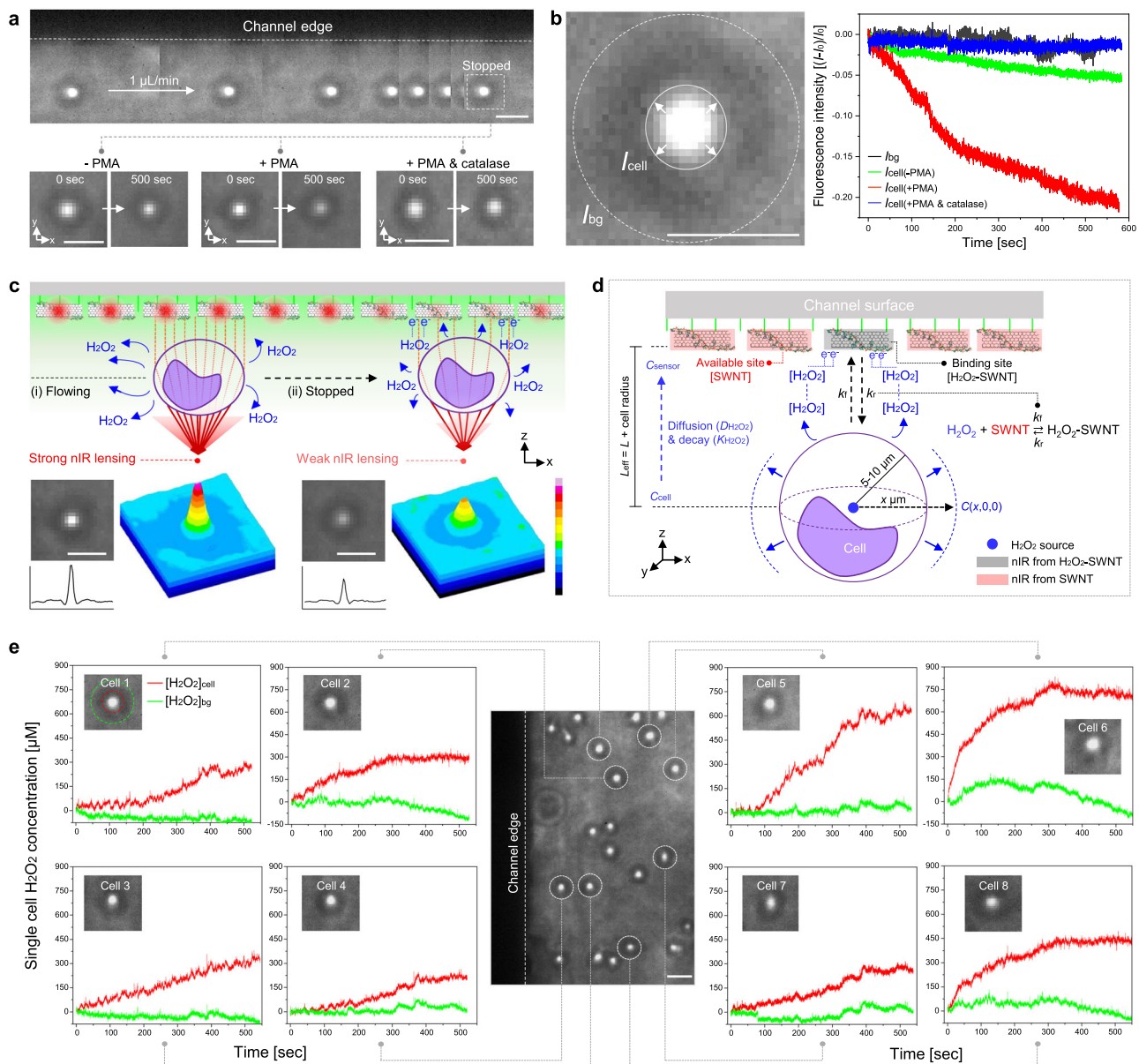

**Fig. 4 Real-time chemical efflux monitoring using the cellular lensing effect. a** Time-series nIR images of a stationary single monocyte with different immune activation states (−PMA, +PMA, +PMA & catalase). **b** Real-time nIR intensity variations of the cells with different activation states. **c** Schematic illustrations of $H_2O_2$ efflux monitoring mechanism with nIR lensing effect. **d** 3D diffusion and reaction kinetic modeling for translation of measured nIR signals to real-time local $H_2O_2$ concentration. **e** Real-time $H_2O_2$ efflux profiles of each single monocyte estimated by the model. 16-color scalebars represent nIR intensity from white (16833) to dark blue (0). Scale bars: 20 μm.

fluorescence resulting from the waveguide light source. We modeled the 3D reaction and diffusion problem of the $H_2O_2$ from the individual cell to translate the observed nIR quenching area above the cell into real-time local $H_2O_2$ concentration (Fig. 4d). An individual cell is assumed to be stationary below the top channel surface with distance $L$ and to instantaneously release $H_2O_2$ molecules at $t = 0$ s. The effective distance between the source and nanosensor array ($L_{eff}$) is then $L_{eff} = L + L_{cell}$, where $L_{cell}$ is the cell radius. The $H_2O_2$ concentration $C$ field is then

$$C(x, y, z, t) = \frac{M}{\left(\sqrt{4\pi Dt}\right)^3} \exp\left(-\frac{x^2 + y^2 + z^2}{4Dt} - Kt\right) \quad (4)$$

where $M$ is the mass flux of $H_2O_2$ release at the cell core, $D$ is the aqueous diffusion coefficient of $H_2O_2$ ($1.5 \times 10^{-5}$ cm$^2$ s$^{-1}$)[73], $K$ is the

first-order decay constant of $H_2O_2$ (from $K = -\ln(0.5)/t_{1/2} = 6.93 \times 10^{-4}$ s$^{-1}$, where $t_{1/2}$ is the cellular half-life of $H_2O_2$ ($10^{-3}$ s))[74] (detail model derivations in Supplementary Note 1). The results show that the $H_2O_2$ efflux reached the nearest nanosensor array quickly at 10 milli sec with a maximum concentration $C_{sensor}$ and the ratio between $C_{sensor}$ and $C_{cell}$ was 0.193 (Supplementary Fig. 20). The adsorption and desorption of $H_2O_2$ on nanosensor array can be described by

$$H_2O_2 + SWNT \rightleftharpoons H_2O_2 - SWNT \quad (5)$$

Corresponding to the rate expression[75]:

$$\frac{d[H_2O_2 - SWNT]}{dt} = k_f[H_2O_2][SWNT] - k_r[H_2O_2 - SWNT] \quad (6)$$

where $k_f$, $k_r$ are the forward and backward rate constants, respectively, and ratio between $k_f$ and $k_r$ was calculated from the

effective equilibrium dissociation constant $K_D = 0.00204$ M. Since the nIR intensity of the nanosensor array is proportional to the fraction of unoccupied sites for binding, [SWNT], or

$$I/I_0 = [SWNT]/[SWNT]_0 \qquad (7)$$

And the number of binding sensor sites are conserved:

$$[SWNT]_0 = [SWNT] + [H_2O_2 - SWNT] \qquad (8)$$

The local concentration of $H_2O_2$ detected by the nanosensor array involves the measured intensity ($I$) and its time-derivative

$$[H_2O_2] = \frac{1}{k_f} \frac{I_0}{I} \left[ k_r \left( 1 - \frac{I}{I_0} \right) - \frac{1}{I_0} \frac{dI}{dt} \right] \qquad (9)$$

Integrating Eq. (9) yields

$$I(t) = I_0/k_s(k_r + k_f[H_2O_2]e^{-k_s t}) \qquad (10)$$

$$k_s = k_r + k_f[H_2O_2] \qquad (11)$$

Equation (10) can be utilized to estimate the real-time local $H_2O_2$ concentration of each single cell from the measured nIR intensity (Fig. 4e). Both the efflux signal ($[H_2O_2]_{cell}$, red line) and background ($[H_2O_2]_{bg}$, green line) for each single monocyte can be measured and differentiated. Furthermore, each monocyte (cell 1–cell 8) demonstrates distinct $H_2O_2$ efflux rates resulting in local concentrations ranging from 175 to 750 μM. This shows that our NCC platform can inform heterogeneities in the efflux rates within cell populations.

**NCC for monitoring of multimodal immune response heterogeneities.** The combination of cellular lensing and label-free nanosensor monitoring within a microfluidic channel allows for real-time chemical efflux cytometry of distinct human monocyte populations, such as those that are immune activated (+PMA) compared to non-activated (−PMA) (Fig. 5a). We show that the NCC platform collects a rich, multivariate data set for each individual cell within the population, that we then easily extract and evaluate with the aid of image analysis code developed as a part of this work (Fig. 5b and Supplementary Fig. 21). The results allow us to plot the real-time $H_2O_2$ efflux rates of two distinct groups ($n = 413$ for −PMA, $n = 414$ for +PMA) versus various key biophysical attributes of each individual cell such as size (cell projected area), eccentricity, and RI (Fig. 5c–e). Since the immune activation of human monocyte is based on PKC-induced mechanistic variations of plasma membrane[76–78], size and eccentricity indicating exterior shape of cell and RI indicating intracellular components variation due to an ion exchange would be key physical variables of monocyte immune responses[79,80]. Therefore, we specifically chose these three biophysical properties to correlate them with main biochemical events (ROS production) to precisely figure out the monocyte response heterogeneity variations during the immune activation process. Upon immune activation, we find that the mean size of monocytes decreases along with a narrowing of the distribution (Fig. 5c). This occurs with an increase in the $H_2O_2$ efflux rate. In contrast, the eccentricity (Fig. 5d) and RI (Fig. 5e) distributions show an insignificant correlation with the $H_2O_2$ efflux rate. To be clear, 3D cytometry and 2D Kernel density estimation show these distinct heterogeneities in detail (Supplementary Figs. 22 and 23, respectively). From these cytometry plots, it is clear that the average $H_2O_2$ efflux rate of activated monocyte population was elevated by 88.9% with a 44.5% larger increase in the variance of the distribution and 30% larger number of high efflux cells compared to non-activated populations (Fig. 5f). The nanosensor array allows us to quantify the mean $H_2O_2$ efflux rates of these two populations as 330 and 624 attomole/cell·min but with $\sigma$ of 344 and 497 attomole/cell·min for −PMA and

+PMA, respectively. In comparison, we measure average values of 59 (−PMA) and 440 (+PMA) attomole/cell·min from the commercial assay Amplex UltraRed kit (Supplementary Fig. 25). The +PMA mean values are in good agreement for the NCC population and commercial assay. However, the mean for the −PMA as measured by NCC is larger than the commercial assay. Further analysis indicates that hyperactive outliers (>1000 atto-mole/cell·min) in this population are the cause of the difference. The mode in the –PMA distribution as measured by NCC of 129 attomole/cell·min is closer to the commercial assay mean, and NCC distribution curves show the significantly higher active tail in Fig. 5f. The ability to detect and quantify this higher producing subpopulation is a clear advantage of NCC over the standard assay. As a consistency check, we note that both methods produce the correct order of magnitude estimate of the $H_2O_2$ efflux rates.

Among the biophysical property changes, the size vs. eccentricity correlation shows the most dramatic change after immune activation (Fig. 5g–i and Supplementary Fig. 24). There is a distinct change in the size distribution upon monocyte activation, with bimodal subpopulations observed for non-activated monocytes with a mean of 271 μm² ($\sigma = 29$) but a single distribution with a lower mean of 263 μm² ($\sigma = 24$) after activation (Fig. 5j). This observation is important because one requires single-cell resolution in order to quantify this type of biophysical change, underscoring an advantage of this NCC platform. Notably, the distributions for both eccentricity (Fig. 5k) and RI (Fig. 5l) remain nearly identical comparing before and after activation but the mean values are slightly shifted from 0.405 ($\sigma = 0.14$) to 0.363 ($\sigma = 0.13$) for eccentricity and 1.383 ($\sigma = 0.05$) to 1.377 ($\sigma = 0.06$) for RI. This indicates that immune activation had a uniform effect on the cell populations with respect to these properties. The ability to detect and analyze subpopulations from a cellular population undergoing biofunctional changes has significant advantages in analytical biochemistry.

Figure 5m summarizes the variation in human monocyte characteristics before and after the immune activation process. The real-time $H_2O_2$ efflux rate of monocyte populations showed 88.9% elevation. Populations showed −2.92% and −10.31% decrease in cell size and eccentricity, respectively, indicating that monocytes appear to shrink and become more circular with immune activation. The RI of the populations decreased by −0.3% scale, which means that light refracted through activated cells produced almost identical refraction angles. This cellular mechanistic insight may lead to additional methods of sorting human monocyte populations. As a consistency check, all of the measured values of NCC were within the ranges previously reported for monocytes, including $H_2O_2$ efflux rate[72]: 100–1000 attomole/cell·min, size[81]: 78.5–314 μm², eccentricity[57]: 0.323–0.473, RI[57,58]: 1.377–1.389. We can safely conclude that our NCC approach is reliable in this way and allows the investigation of multiple cellular parameters of a given population in real-time and at high throughput. These cellular physical parameter changes of monocytes during activation are consistent with PKC translocation effects. It is known that when monocytes are activated by PMA, PKC proteins are translocated from the cytosol to the plasma membrane, activating NADPH oxidase with an increase in ROS generation[76]. Subsequently, fluidity and permeability of the cellular membrane are both downregulated upon PKC integration[77,78]. One expects a resulting decrease in monocyte deformability[79], with cells becoming more rigid, smaller and circular consistent with our observations in this work. In addition, the concentration of intracellular solutes is lower than normal since ion exchange and solution transport are hindered upon PKC integration with lower membrane fluidity[80]. This is consistent with the slightly lower observed RI per cell. We note that a single

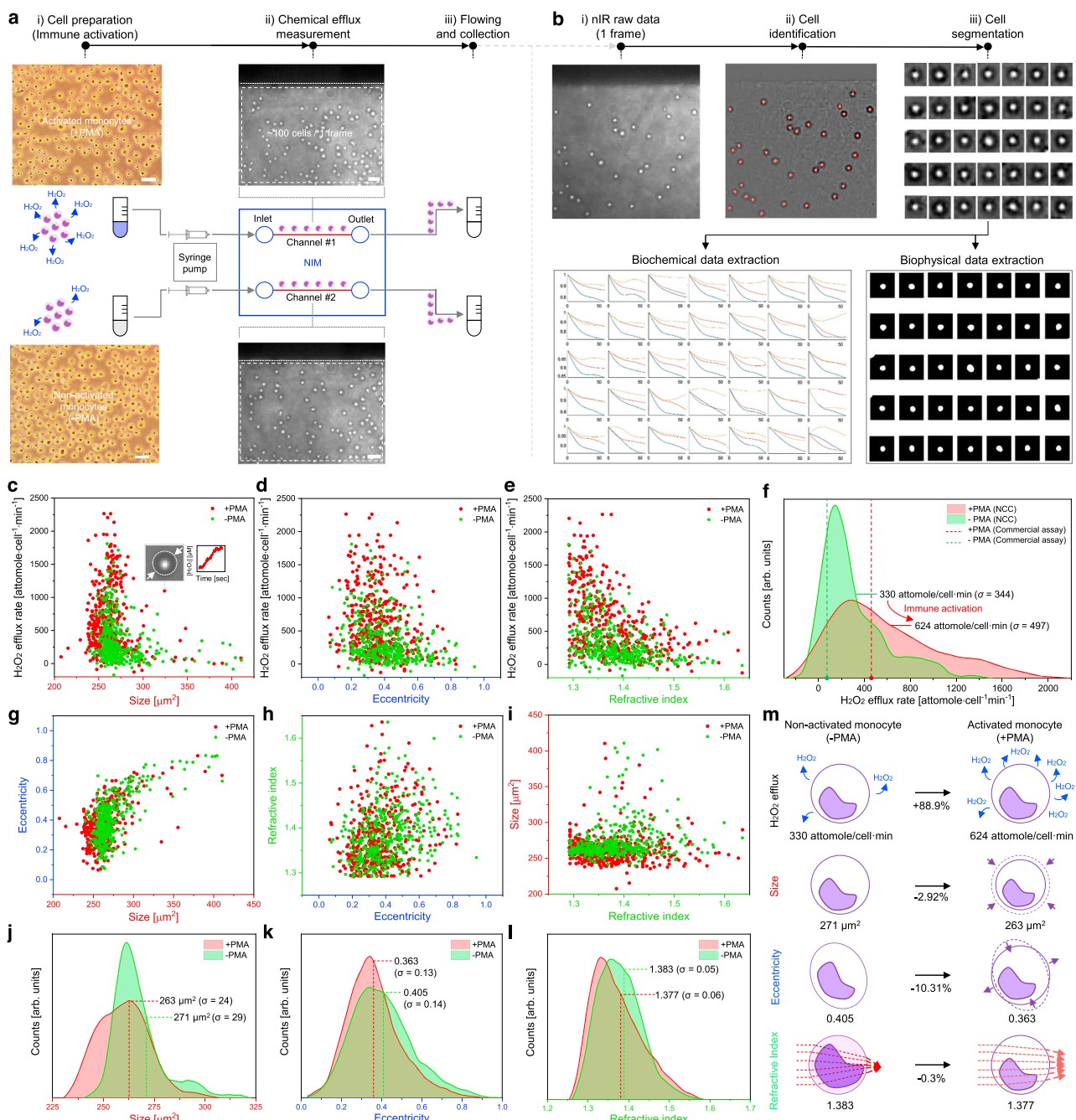

**Fig. 5 NCC for monitoring of multimodal immune response heterogeneities. a** Schematics and nIR images of NCC set up with distinct activation of human monocytes (−PMA and +PMA). **b** Automatic nIR image analysis using computational code for cell data extractions. **c**–**e** NCC cytometry plots of $H_2O_2$ efflux rate vs. biophysical parameters ((**c**) size (2D projected area), (**d**) eccentricity, (**e**) RI) of two monocytes populations. Data are $n_{cell} = 413$ for −PMA, $n_{cell} = 414$ for +PMA from $n = 6$ biologically independent samples. **f** NCC distribution curves of $H_2O_2$ efflux rates with data from commercial assay kit. **g**–**i** NCC cytometry plots for cell biophysical parameters ((**g**) eccentricity vs. size, (**h**) RI vs. eccentricity, (**i**) size vs RI). **j**–**l** NCC distribution curves of each biophysical parameters ((**j**) size, (**k**) eccentricity, (**l**) RI). **m** Schematics illustrations for cell properties variations of human monocyte populations with immune activations. Scale bars: 20 μm.

platform that can track these complex functional changes at the single-cell level is a substantial advance over the state of the art.

The monitoring throughput of our platform can be simply determined as a total number of cells in one frame of nIR image divided by whole processing time (data acquisition time + efflux detection time (10 min)) (Supplementary Fig. 26). Then, experimental throughput of the NCC technique for monocyte $H_2O_2$ efflux monitoring is at an estimated ~600 cells/h for

single-channel operation with real-time single cell resolution. This appears to be at the highest range demonstrated to date for conventional chemical cytometry with reported values are 50–500 cells/h[27–29]. The previous reports also excluded labeling process time (usually taking a few hours) based on only the data acquisition time[24]. Our next steps for NCC will utilize a parallel and multiple channel approach, automated fluidic control coupled to automated data collection to increase this throughput

substantially. Preliminary work showed that analyzing cell number could be improved with 300% increment based on identical processing time by using three parallel channels of integrated microfluidic chip, achieving ~1800 cells/h throughput (Supplementary Fig. 27). We note that the existing process reported in this work appears gentle enough to maintain the cell viability. We verified this for monocyte populations after a series of NCC experiments (Supplementary Fig. 28). The viabilities of measured cell populations were statistically identical to reference populations (unprocessed) indicating that our NCC platform can be used for completely non-destructive single-cell analysis with potential for cellular therapeutics.

In conclusion, we demonstrate a non-destructive chemical cytometry that integrates nanosensor arrays within a cell-transporting microfluidic channel, capable of exploiting cellular lensing for high-resolution detection of single-cell chemical efflux. Using this platform, large number of single cells can be imaged and analyzed on the array via their projected nIR lensing image, producing a profile that matches the predictions of photonic nanojet model. The result is a unique tool capable of multimodal biophysical characterization of individual cells, including their size, eccentricity, and RI, all at high throughput. With this biophotonic waveguide, the chemical efflux of single-cell was label-free monitored in real-time at the attomolar level. We use this NCC tool to study the heterogeneity of the immune response of distinct human monocyte populations at the highest throughput range for chemical cytometry in a completely non-destructive manner. Mathematical analysis of the resulting rich data sets reveals distinct phenotypic correlations between chemical efflux and biophysical properties that can be quantified and used to understand cellular biochemistry and mechanistic pathways. For example, we find that real-time $H_2O_2$ efflux of human monocytes is unusually heterogeneous and distinctly related to biophysical parameters following immune activation. The measured $H_2O_2$ efflux rates between 330 and 624 attomole/cell min corresponded to overall cell size ranges of 271 and 263 $\mu m^2$, eccentricity values between 0.405 and 0.363 and RI values between 1.383 and 1.377 for non-activated and activated monocytes, respectively. Thus, we highlight that NCC is able to profile immune cell heterogeneities allowing for monitoring of variances in cell therapeutics. We also demonstrate the ability to incorporate sensors for multiple molecular targets of cells. Our platform is label-free and uses the unique property of cellular lensing to extract molecular signals on a population scale. We believe that the NCC platform can be readily extended to various biochemical efflux monitoring of cell types such as neurons, cancer cells or stem cells given the appropriate choices of sensor–analyte pairs (Supplementary Fig. 29). We envision that our nanotechnology-based biophotonic cytometry provides a unique strategy for coupling nanosensors into a form-factor that enables single-cell analysis of relevant populations for cellular manufacturing, cellular immunology, and biopharmaceutical research.

## Methods

**Preparation and characterization of nanosensors**. HiPco[TM] SWNTs purchased from Unidym were suspended with a 30-base (GT) sequence of ssDNA (Integrated DNA Technologies) in a 2:1 DNA:SWNT mass ratio in 0.1 M NaCl solution. (ATCAAGGCTCGAATTGTCCCTGAAATCT) sequence was used for random DNA and PS sulfonate/bromostyrene was used for random copolymer in reference test. A typical DNA concentration was 2 mg/mL. Samples were sonicated with a 3 mm probe tip (Cole Parmer) for 10 min at a power of 10 W and 40% amplitude in an ice bath. Then samples were centrifuged twice for 90 min (Eppendorf Centrifuge 5415D) at 16100 relative centrifugal force (RCF). Afterwards, the supernatant was collected and the pellet was discarded. UV–Vis–nIR absorption spectra

(Cary 5000, Agilent Technologies, Inc.) were collected to verify successful suspension of nanosensor. Nanosensor concentration in the dispersion was estimated using an extinction coefficient of $\varepsilon_{632 \, nm} = 0.036 \, (mg/L)^{-1}$. Final concentration of SWNT/(GT)$_{15}$ is from 10 to 80 mg/L. 80 mg/L concentration of nanosensor dispersion was used to all NIM experiments.

**Nanosensors integration with microfluidic channel**. Microfluidic channels (detail specification in Supplementary Table 1) were purchased from ibidi[R] (μ-Slide VI 0.1, ibiTreat). 2 μL of APTES (99%, Sigma Aldrich) in ethanol (1% APTES, 1% $H_2O$) was injected to microfluidic channel with micro-pipetting and treated for 3 h. After APTES treatment, 2 μL of nanosensor dispersions were injected. After overnight evaporation, SWNT/(GT)$_{15}$-coated channel surfaces were rinsed with 1 mL 1× PBS (pH 7.4, Life Technologies[TM]) twice to remove the unbounded nanosensor. 0.8 mm Silicone tubes were connected with NIM using Elbow Luer Connector Male (ibidi[R]). In vitro $H_2O_2$ detection experiments were conducted as below. SWNT/(GT)$_{15}$ releases the nIR fluorescence with visible range excitation laser (e.g. 516 nm) acting as an optical transducer for $H_2O_2$ detection. Aqueous $H_2O_2$ solution (30 wt%, Sigma Aldrich) was diluted with distilled $H_2O$ from 1 μM to 1 M to investigate chemical sensing performance of NIM. Diluted $H_2O_2$ solutions were flowing through the NIM with syringe pump (0–1 μL/min, Harvard Apparatus) and averaged quenching signals from nanosensor array (250 × 350 μm²) were recorded for 500–600 s. Recorded nIR images were processed by ImageJ with Gray and Fire scales to clearly visualize the variations of nIR intensities.

**Characterization and nIR measurements**. Raman spectroscopy (Horiba Jobin Yvon LabRAM HR800) was used to investigate the nanosensor assembly direction in NIM with a 532 nm laser excitation (3 s accumulations) and ∼1 μm of spot size with 1800 lines/mm grating. The $G$ band originating from tangential oscillations of the carbon atoms in the SWNT was observed in the frequency range of 1590 cm$^{-1}$. When $\theta = 0°$ and $\theta = 90°$, the incident excitation polarization direction was parallel and perpendicular to the flowing direction of the microfluidic channel, respectively, indicating that the SWNT/(GT)$_{15}$ nanosensors were aligned along the flowing direction of channel during EISA. AFM profiles of nanosensor array were scanned with Bruker Multimode 8 with Controller V. AFM images were taken in the ScanAsyst tapping mode in the air with TESPA probes having an elastic constant of 42 N/m and tip radius of 8 nm. The images were recorded with the scan rate of 1 Hz and a resolution of 1024 lines per image for each area respectively, recorded at three different places of the single-channel surface. Image analysis was done with NanoScope Analysis software 1.4 from Bruker. nIR spectrum of NIM were collected with a fluorescence spectrometer equipped with a 785 nm photodiode laser (B&W Tek. Inc. 450 mW). Low-magnified nIR images were collected with a Zeiss AxioVision inverted microscope with appropriate optical filters. The fluorescence passed through an Acton SP2500 spectrometer (Princeton Instruments), and measured with a liquid nitrogen cooled InGaAs 1D detector (Princeton Experiments). Inverted OM (Eclipse TS100, Nikon) was used for NIM and flowing cell imaging with visible light. NCC were implemented and recorded by nIR microscopy hyperspectral imager (IMA IR[TM], Photon Etc.). NCC was implemented with the help of a nIR microscope (IMA IR[TM], Photon Etc.) equipped with 561 nm laser excitation (MGL-FN-561, Opto Engine LLC). The laser power was adjusted from 30 to 350 mW with optical density filters (laser power control in Supplementary Fig. 8). The laser was passed through a laser line filter, reflected by dichroic mirror, and focused onto the back focal plane of an inverted objective to illuminate the entire field of view of the NIM under study. nIR fluorescence from the NIM passed a longpass filter and was measured using a TE cooled infrared camera. All the measurements were conducted with ×20 objective, 0.1 s exposure time and medium intensity gain. In order to investigate the focal points and observed cell locations, motorized Z-stage controller was integrated with nIR microscopy. Hollow glass microspheres (0.6 g/cc and 5–30 μm, Cospheric LLC), PS microparticle (20 μm, Sigma Aldrich), and stainless steel metal microspheres (7.8 g/cc and 1–22 μm, Cospheric LLC) were used for reference particles as lensing effect observations. All reported micrograph results were consistently replicated across multiple experiments (minimum of $n = 3$) with all replicates generating similar results.

**FDTD numerical modeling**. FDTD modeling for nIR photonic nanojet were performed using Lumerical FDTD Solution (Lumerical Inc). Micro-spherical structures having various range of size (radius: 1, 2, 3, 4, 5, 6, 7, and 8 μm), eccentricity (z-axis distance: 2.5, 3, 3.5, 4, and 4.5 μm), and RI ($n_c/n_m$: 1.01, 1.02, 1.03, 1.04, 1.05, 1.06, 1.07, 1.08, 1.09, and 1.10) were set and excited by an incident plane wave with a various range of wavelength ($\lambda = 550$, 650, 700, 750, 800, 1000 nm) including fluorescence emission of the nanosensor array. The calculation domain was $50 \times 50 \times 50 \, \mu m^3$ and uniform mesh of around 30 nm was used. The perfectly matched layers (PML) were arranged around the boundaries. RI of media (out of cell) was set to 1.33.

**Cell experiments**. Monocytes (U937, ATCC CRL-1593.2), B lymphocytes (FIB504.64, ATCC HB-293), epithelial (HEK-293, ATCC CRL-1573), and endothelial (HUVEC, ATCC CRL-1730) cells were purchased from American Type

Culture Collection (ATCC) and cultivated according to the supplier's protocol. U937 and FIB504.64 were cultured in RPMI-1640 (ATCC 30-2001) with 10% of fetal bovine serum (FBS) (A3160601, Gibco$^{TM}$). HEK-293 cells were cultured in Dulbecco's modified essential medium (DMEM; Lonza) with 10% FBS (ATCC 30-2020). HUVEC were cultured in F-12K medium supplemented with 10% FBS (ATCC 30-2020), 1% endothelial cell growth factor (100×, Sigma), 100 IU/mL penicillin, and 100 μg/mL streptomycin. For the adherent HUVEC observations, microfluidic channels were initially coated with endothelial cell attachment factor (ECAF) to promote HUVEC cell adherence on channel surfaces. All the cells were cultured in 75 cm$^2$ cell culture flasks (Falcon) under incubating conditions of 5% CO$_2$ at 37 °C (Forma™ series II 3110, ThermoFisher Scientific). Three days of cultured U397 were used (cell number: $10^4$–$10^5$/mL, passage number = 4) to implement NCC in this study. To monitor only the instantaneous H$_2$O$_2$ efflux, cell media was changed by fresh PBS with 10 min 130 RCF centrifugation at 10 °C so that remove all the by-product, accumulated efflux and abnormal cells in media. 10 μL of 0.5 mg/mL PMA (Sigma Aldrich, for use in molecular biology applications, ≥99%) was added in 1 mL of U937 cell media to activate the monocyte and induce differentiation into macrophage (final concentration of PMA = 5 μg/mL). 100 mL of 200 units/mL Catalase (Sigma Aldrich, from bovine liver) was used for H$_2$O$_2$ removal control experiments. For final NCC implementation, activated (+PMA) and non-activated monocytes (−PMA) were flowing through NIM using syringe pump (Harvard Apparatus) with flowing rate from 0 to 10 μL/min. PMA group was firstly injected through channel 1 with syringe pump and measured at the stopped position for 10 min to accumulate the H$_2$O$_2$ efflux on nanosensor array. Then, measured cells were flowed (10 μL/min) and collected in an empty tube for future experiments. Lastly, +PMA group was injected to channel 2 and H$_2$O$_2$ efflux was measured. NCC was conducted for stationary cells for few min and videos were recorded to analyze efflux signals of the cells. Attomolar efflux rates were calculated from real-time local H$_2$O$_2$ concentration multiplied with single unit volume (single monocyte volume = $4.18 \times 10^{-15}$ m$^3$) and divided by measurement time (10 min). Six biological replicates of U937 populations were used for NCC cytometry plot data.

**Data analysis**. nIR image analysis and quantitation was performed in MATLAB (Natick, MA) with the steps detailed below. Cell identification is performed by taking 1 frame of the nIR video (500 s recorded, 0.1 s of exposure time, 5000 frames) convolving with a Laplacian of a Gaussian filter, and then thresholded by the user for each experiment batch. For each cell, the image is then interpolated. Using the peak and Airy ring of the nIR lensing spot, the cell image is normalized, and then statistics such as cell size and eccentricity are evaluated with Regionprops function. The projected area (i.e. size) values are dilated appropriately to coincide with the photonic nanojet model. To avoid excess data interpolation, camera pixel intensities are used for subsequent analysis. The cellular lensing intensity ($I_0$) is found by choosing the camera pixel closest to the centroid. To calculate background, 16 pixels outside of the secondary peak of the lensing effect is chosen. Outliers are then removed, and background traces are averaged to use as normalization for the centroid intensity traces.

**Reporting summary**. Further information on research design is available in the Nature Research Reporting Summary linked to this article.

## Data availability
The authors declare that all data supporting the findings of this study are available within the paper and any raw data can be obtained from the corresponding author on request.

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

## Acknowledgements

The authors are grateful for financial support from Bose Fellowship Award to M. S. S., the Juvenile Diabetes Research Foundation (JDRF), funding from the Disruptive & Sustainable Technology for Agricultural Precision (DiSTAP) and the Singapore MIT Alliance for Research and Technology (SMART) Center, and the Walmart Foundation and the Walmart Food Safety Collaboration Center in Beijing. T. T. S. L. acknowledges a graduate fellowship by the Agency of Science, Research and Technology, Singapore. M. K. acknowledges support by the German Research Foundation (DFG) Research Fellowship KU 3952/1-1. We give thanks to the Nanotechnology Materials Lab, and the Koch Institute for Integrative Cancer Research, MIT for AFM measurements.

## Author contributions

S.-Y. C. and M.S.S. conceived the idea, designed the project and planned experiments with the assistance of X.G., V.B.K., M.K., S.J.M., M.S., T.T.S.L., P.G., X.J. and S.-Y.C. prepared the nanosensors, fabricated the NIM, implemented the NCC with cell culturing, measured and analyzed the data. X.G. coded automatic image analysis program and conducted cell data processing. V.B.K. conducted FDTD numerical modeling for photonic nanojet demonstration. M.K. assisted nIR observations of NIM. S.J.M. assisted HEK cell culturing and commercial H2O2 assay. M.S. assisted with polarized Raman spectroscopy and nanosensor synthesis. T.T.S.L. commented on H2O2 nanosensor and their detection mechanism. P.G. assisted with AFM characterization of nanosensor array. X.J. assisted with preparation of various nanosensor. S.-Y.C. and M.S.S. wrote the manuscript with inputs from all the authors. All authors contributed to discussions informing the research.

## Competing interests

The authors declare no competing interests.
