## [Peer Review File · Nature Communications]

REVIEWERS' COMMENTS

Reviewer #1 (Remarks to the Author):

In this manuscript, the authors proposed and experimentally demonstrated a chemical flux cytometry method that simultaneously measured a few biophysical properties. These were enabled by a combination of a carbon nanotube-based nanosensor array and a cellular lensing effect. Although the authors have previously reported the nanosensors, I recognize the novelty of this work; in particular, the simultaneous detection of chemical and biophysical properties is unique. This work was originally submitted to another journal, for which I reviewed it. I did not think it was publishable in the journal, but do think this revised version is acceptable at the level of *Nature Communications*. I have one minor comment to reinforce the significance of the authors' argument; the authors should add a few references to the sentence "Single the immune activation ... of monocyte immune responses" in the section titled "NCC for Monitoring ...". If the authors address this point, I would be happy to recommend publication of the work in *Nature Communications*.

Response to Referee #1

Comment: In this manuscript, the authors proposed and experimentally demonstrated a chemical flux cytometry method that simultaneously measured a few biophysical properties. These were enabled by a combination of a carbon nanotube-based nanosensor array and a cellular lensing effect. Although the authors have previously reported the nanosensors, I recognize the novelty of this work; in particular, the simultaneous detection of chemical and biophysical properties is unique. This work was originally submitted to another journal, for which I reviewed it. I did not think it was publishable in the journal, but do think this revised version is acceptable at the level of *Nature Communications*. I have one minor comment to reinforce the significance of the authors' argument; the authors should add a few references to the sentence "*Since the immune activation ... of monocyte immune responses*" in the section titled "*NCC for Monitoring ...*" If the authors address this point, I would be happy to recommend publication of the work in *Nature Communications*.

Response: We thank the reviewer for the positive comments regarding novelty of our work and for the valuable feedback during the previous round of reviews. As the reviewer suggested, we added few references on the sentence as below.

(page 15-16) "*Since the immune activation of human monocyte is based on PKC induced mechanistic variations of plasma membrane,⁷⁷⁻⁷⁹ size & eccentricity indicating exterior shape of cell and RI indicating intracellular components variation due to an ion exchange would be key physical variables of monocyte immune responses.⁸⁰⁻⁸¹*"